# Markovian Embeddings for Coalitional Bargaining Games

**Lucia Cipolina-Kun**
School of Computer Science, Electrical and Electronic and Engineering Maths
University of Bristol. UK
`lucia.kun@bristol.ac.uk`

## Abstract

We examine the Markovian properties of coalition bargaining games, in particular, the case where past rejected proposals cannot be repeated. We propose a Markovian embedding with filtrations to render the sates Markovian and thus, fit into the framework of stochastic games.

## 1 Introduction and Related Literature

Coalitional bargaining games CBG are sequential games where one agent at random proposes a coalition formation while the others provide responses on whether to accept or reject the proposal. If a proposal is accepted, the game terminates and the coalition is formed [1]. If a proposal is rejected, the game continues with another proposer at random formulating a different coalition proposal.The goal is to find an agreement over the coalition members. A CBG can we framed as a stochastic game where the game states are configured by two sequential actions: the proposals over coalition members and the corresponding responses. The state dynamics are determined by the agent's preferences and the rewards are assigned once a proposed coalition is accepted. The order of coalition proposals is a crucial factor in determining the convergence of CBG Okada (1996); however, it has not received sufficient attention in the existing literature Rubinstein (1982); Okada (1996); Chatterjee et al. (1993) and a thorough analysis is missing. Given that the speed of convergence of the game can vary significantly depending on whether proposals can be repeated, a comprehensive analysis of this aspect is of great importance. Specifically, the repetition of proposals can result in three different scenarios regarding the speed in which an agreement is reached. First, if rejected proposals are allowed to be repeated in future rounds (implying that agents may reconsider their options later) the CBG process turns into a multi-dimensional random walk bouncing back and forth between states delaying agreement. A second scenario like in Bachrach et al. (2020) allows for repetition of proposals but introduces learning agents *learn* avoiding the repetition of proposals through a reward signal. In this case, the learning aspect converts the CBG process from a pure random walk to a stochastic game with learned transition dynamics. A third scenario is to restrict the proposals of coalitions already rejected in the past. This is the most natural setting and allows the CGB to converge to an agreement in the most efficient way.

The three scenarios have different implications on the Markovian property of the CBG process. This property requires that the transition probability between states depends solely on the current state and action, regardless of the past trajectory of states. This is the case on the first and second scenarios described above; however, on the third scenario, where rejected proposals can no longer be proposed in the future, the CBG process is no longer Markovian, as the probability distribution of the next proposal depends on the entire history (i.e., past proposals) of the game. The Markovian property of coalitional bargaining games is relevant when using multi-agent reinforcement learning (MARL) to approximate the optimal policies of agents in the game. MARL is based on the theoretical framework of stochastic games, which require the transition dynamics between states to be Markovian Littman (1994). Solution concepts commonly used in stochastic games such as the Markov perfect Nash equilibrium Yang & Wang (2020), require the Markovian property to hold. Thus, understanding the Markovian property of CBG is crucial in designing effective multi-agent reinforcement learning algorithms for optimizing agent's performance.

---

[1] To simplify and without loss of generality, we leave the coalition payoff aside.

**Our contributions.** We examine the most natural case of CBG in which proposals, once rejected, cannot be repeated. First, we provide a proof that such a game is non-Markovian. Second, we present a solution to convert it into a Markovian game by proposing a *Markovian embedding* with *filtrations*. As a result, the resulting Markovian game captures the same information as the original non-Markovian game, thus allowing the application of MARL frameworks.

## 2 CBG AS A NON-MARKOVIAN STOCHASTIC GAME

Consider a CBG involving a set of agents denoted by $N = \{1, 2, \ldots, n\}$ and a set of possible coalitions denoted by $c \subseteq 2^N$. At time $t = 0$, a proposer $i$ is chosen uniformly at random from $N$, and proposes a coalition $c_0 \subseteq N$ to the other agents, on the same time step, the agents in $c_0$ reply "accept" or "reject" in turns. If a proposal is accepted, a coalition is formed and the game terminates; otherwise, if a proposal is rejected, the game proceeds to time $t = 1$, and a new proposer is chosen uniformly at random from the agents who have not yet proposed and proposes a new coalition $c_1 \subseteq N$. Again, the agents in $c_1$ reply "accept" or "reject" in turns, and the game proceeds in this way until either a proposal is accepted, or all possible coalitions have been proposed and rejected. Consider the state of the game at time $t$ as defined by Okada (1996); Chatterjee et al. (1993) $s_t = (p_t, c_t)$, where $p_t$ is the current proposer and $c_t$ is the proposed coalition. Defined like this, the current state only contains information on the current proposal; however, the probability of the next state $P(s_{t+1})$ depends on the set of rejected proposals up to time $t$, and not just the current state. As such, since the events $s_{t+1}$ and $(s_{t-1})_{t>0}$ are not independent of each other and we have:

$$P(s_{t+1}|s_0, s_1, \ldots, s_t) \neq P(s_{t+1}|s_t) \tag{1}$$

## 3 MARKOVIAN EMBEDDING OF THE COALITION BARGAINING GAME

In the previous section, we showed that the CBG with the restriction on repeating proposals is non-Markovian. However, we can convert this non-Markovian process into a Markovian one by introducing a *Markovian embedding* using a *filtration*. In this section, we will show how this can be done. A Markovian embedding with *filtrations* is a probability space $(\Omega, \mathcal{F}, P)$ equipped with a sequence of sub-sigma-algebras $(\mathcal{F}_t)_{t>0}$, where $\mathcal{F}_t \subseteq \mathcal{F}$ captures the *ordered* history of the game up to time $t$ including proposals, acceptances, and rejections i.e., $\mathcal{F}_t = \sigma\big((i_1, o_1), \ldots, (i_k, o_k) \mid 1 \leq i_j < j, 1 \leq j \leq k \text{ and } k \leq t\big)$, where $(i_j, o_j)$ denotes the outcome of the $j$th proposal, with $i_j$ being the proposer and $o_j$ being the outcome (either accepted or rejected).

Let's now define a new state of the game as $s_t = (c_t, p_t, \mathcal{F}_t)$. This sate captures all the relevant information needed to determine the future behavior of the game. Specifically, the next state is obtained by updating the set of proposals $c_t$ based on the action taken and updating the filtration $\mathcal{F}_t$ based on the outcome of the action. The new state is an *adapted* stochastic process $(s_t)_{t>0}$ defined on this probability space, such that the Markov property holds. In other words, the conditional distribution of $s_{t+1}$ given $\mathcal{F}_t$ depends only on $s_t$ and not on any earlier values of the process. With this definition, we can show that the Markov property holds, as follows:

$$P(s_{t+1} \mid \mathcal{F}_t) = P(s_{t+1} \mid s_t) \tag{2}$$

The above Equation 2 holds since the filtration is a sequence of *nested* sigma-algebras. Hence, the conditional probability given all the sigma-algebras is the same as the conditional probability given the last one in the sequence. A longer proof can be found on Appendix 5.5.

## 4 CONCLUSIONS AND FUTURE WORK

We have analyzed the implications of different state definitions on a CBG showing that while is natural to avoid repetition of proposals to improve convergence, this can render the game non-Markovian, making it difficult to apply MARL/stochastic game results. We have also shown how to embed the non-Markovian process into a Markovian one using a filtration.

URM STATEMENT

The authors acknowledge that Lucia Cipolina-Kun meets the URM criteria of the ICLR 2023 Tiny Papers Track.

ACKNOWLEDGEMENTS

This work was supported by the UK Engineering and Physical Sciences Research Council (EPSRC) through a Turing AI Fellowship (EP/V022067/1) on Citizen-Centric AI Systems. (https://ccais.soton.ac.uk/). Lucia Cipolina-Kun is funded by British Telecom.

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

## 5 APPENDIX

This section provides further definitions and contextual information relevant to the content of the paper.

### 5.1 STOCHASTIC GAMES

A stochastic game (Shapley, 1953) generalises Markov Decision Processes to involve multiple agents, being the reason why they are the mathematical framework for MARL [2]. The idea behind a

---

[2]MARL is a useful computational framework for stochastic games when the transition dynamics are unknown.

stochastic game is that the history at each period can be summarized by a *state*. Stochastic games are defined as a tuple $<N, S, A, T, R, \gamma>$ where: $N$ denotes the set of n agents, $S$ denotes the set of states, $A = A_i \ldots A_n$ denotes the set of joint actions, where $A_i$ is player $i's$ set of actions. $T : S \times A \to S$ denotes the transition dynamics, $R : S \times A \times S \times N \to R$ denotes the reward function and $\gamma$ denotes the discount factor.

The image below depicts the relationship between the different categories of stochastic games and CBG.

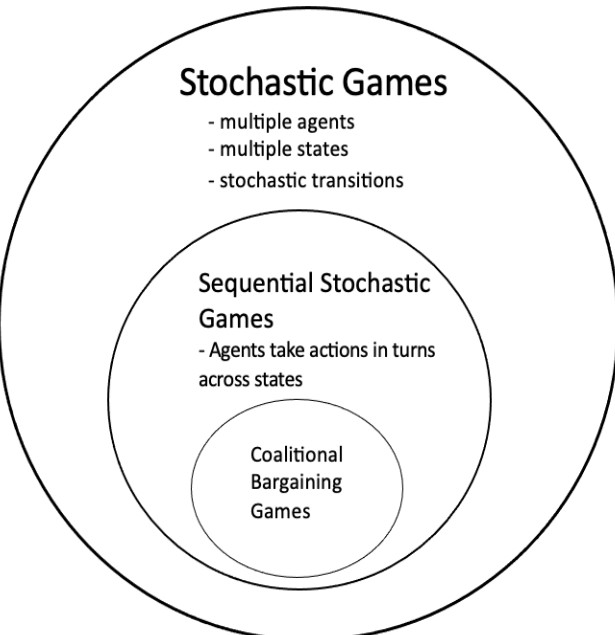

Figure 1: Relationship between stochastic games, sequential stochastic games and coalitional bargaining games.

## 5.2 THE COALITION STRUCTURE GENERATION PROBLEM

The Coalition Structure Generation (CSG) problem is the process of forming coalitions among agents in such a way that the agents within each coalition coordinate their activities, but there is no coordination between coalitions. This entails dividing the set of agents into exhaustive and mutually exclusive coalitions. The resulting partition is referred to as a Coalition Structure (CS). After defining the optimal Coalition Structure, a new game starts to divide the value of the generated solution among agents.

## 5.3 NON-COOPERATIVE BARGAINING THEORY OF COALITION GENERATION

The non-cooperative game approach to the problem of cooperation was initiated in the seminal works of Nash (1950; 1953), who presented equilibrium results for finite-horizon two-person bargaining game known as the Nash Program. The approach aims to explain cooperation as the result of individual players' payoff maximization in an equilibrium of a non-cooperative bargaining game that models pre-play negotiations. Nash stated the seminal results that cooperation should be strategically stable. The approach re-examines a widely held view in economics, called the efficiency principle, that a Pareto-efficient allocation of resources can be attained through voluntary bargaining by rational agents if there is neither private information nor bargaining costs. After Nash, the theory centered its attention into extending the result to infinite-horizon bargaining. The work of Rubinstein (1982) introduces the *alternating offers model* as an equilibrium bargaining protocol for two-person infinite-horizon bargaining. The expansion of this model to n-person bargaining came later with the work of several authors (see Chalkiadakis et al. (2011) for a literature review). One

example is the protocol proposed by Okada (1996), which presents a sequential bargaining game in which players propose coalitions and feasible payoff allocations until an agreement is reached. Under this protocol, agreement can be reached in one bargaining round if the proposer is chosen randomly.

The image below depicts the stages of a coalitional bargaining game.

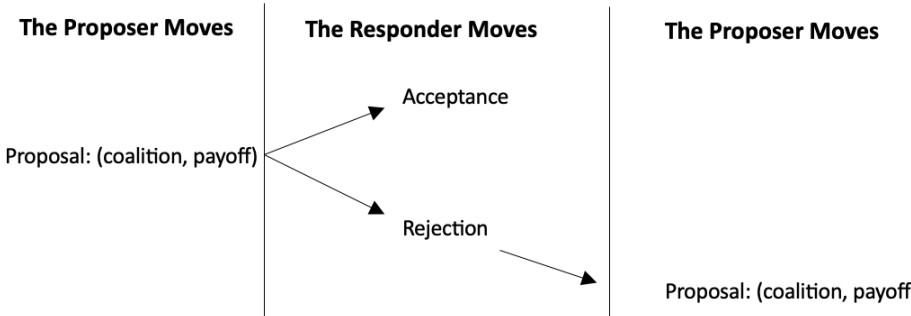

Figure 2: Bargaining game as an extensive-form game.

### 5.4 PROBABILISTIC PROPERTIES OF COALITIONAL BARGAINING GAMES

This section expands on probability concepts related to coalitional bargaining games.

**The stationarity property of policies in coalitional bargaining games.** In CBG, the rewards received by each agent depend on the state of the negotiation, which is determined by the actions taken by all agents in *previous* rounds of negotiation. As agents learn to negotiate and adjust their policies, the environment dynamics reflects these changes.

**Introducing sigma-algebras and filtrations on the state definition.**

Formally, in a coalitional bargaining game, the **sigma-algebras** represent the information available to each player at each stage of the game. Specifically, the sigma-algebra $\sigma_t$ at time $t$ is defined as the collection of variables including on the game's state space that a player knows with certainty at that time. This includes all of the player's own previous actions, as well as any public information (such as the offers made by other players) that the player has observed up to that point.

A **filtration** is an *ordered* sequence of sigma-algebras that captures the increasing amount of information available to a player as the game progresses. Formally, a filtration on a probability space $(\Omega, \mathcal{F}, \mathbb{P})$ is a sequence of sigma-algebras $\mathcal{F}_{t \geq 0}$ such that $\mathcal{F}_0$ is the trivial sigma-algebra and $\mathcal{F}_t \subseteq \mathcal{F}_{t+1}$ for all $t \geq 0$.

**Markovian embeddings**. A non-Markovian game such as coalitional bargaining can be converted into a Markovian game by defining a Markovian embedding on the state with filtrations. In a non-Markovian game, the future state of the game depends on the entire history of the game up to that point; however, by adding the filtration and the sigma-algebras in the state definition, we can reduce the game to a Markovian form. Specifically, at each stage of the game, the player's action depends only on the information available in the corresponding sigma-algebra. By using the filtration to accumulate the sigma-algebras, we can capture the player's increasing information and define the game in a Markovian form.

**Relationship between filtrations and Markov property.** Filtrations have a natural relationship with Markov process. By definition, in a Markov process, conditioning on a filtration gives the same result as not conditioning on it because the future state of the process depends only on its current state, and not on its past history. In coalitional bargaining, we can include filtrations in the definition of the environment's states so that the Markovian property holds. The filtration is simply a way of summarizing the past history up to a certain point, including information such as the current offers, counter-offers, and agreements made by the agents, as well as any other relevant information about the negotiation. By adding the filtration to the current state, the future state of the process

depends only on the current state (and the agent's action), and thus it now does not matter whether we condition on the filtration or not.

## 5.5 LONG PROOF OF EQUATION 2

Recall that we have defined a Markovian embedding as a stochastic process $\{s_t, \mathcal{F}_t\}_{t \geq 0}$ where $s_t$ is a random variable representing the state of the system at time $t$, and $\mathcal{F}_t$ is a filtration that captures the information available up to time $t$ (i.e., the sequence of proposal outcomes up to time $t$.).

The Markov property requires that the future state of the system at time $t + 1$ depends only on the present state $s_t$ and not on the entire history of the system up to time $t$ beyond $s_t$. We will prove that:

$$P(s_{t+1}|\mathcal{F}_t, s_t) = P(s_{t+1}|s_t) \tag{3}$$

This means that given the current state $s_t$ and the information up to time $t$ captured by $\mathcal{F}_t$, the conditional probability distribution of the future state $s_{t+1}$ is the same as the unconditional probability distribution of $s_{t+1}$ given only $s_t$.

We can rewrite the left-hand side of the above equation using the definition of conditional probability as:

$$P(s_{t+1}|s_t, \mathcal{F}t) = \frac{P(s_{t+1}, s_t|\mathcal{F}_t)}{P(s_t|\mathcal{F}_t)} \tag{4}$$

By the definition of conditional probability, we know that $P(s_{t+1}, s_t \mid \mathcal{F}_t) = P(s_{t+1}, s_t)$. To see this, start from $P(s_{t+1}, s_t \mid \mathcal{F}_t)$ and apply the definition of conditional probability:

$$P(s_{t+1}, s_t|\mathcal{F}t) = \frac{P(s_{t+1}, s_t)P(\mathcal{F}_t)}{P(\mathcal{F}_t)} = P(s_{t+1}, s_t) \tag{5}$$

Here, we have used the fact that the denominator $P(\mathcal{F}_t)$ cancels out. Also, note that $P(s_t|\mathcal{F}_t)$ depends only on the state $s_t$ and not on any past history beyond $s_t$. Therefore, we can rewrite Equation 4 as:

$$P(s_{t+1}|s_t, \mathcal{F}_t) = \frac{P(s_{t+1}, s_t)}{P(s_t)} \tag{6}$$

Now, we want to show that the above equation is equivalent to $P(s_{t+1}|s_t)$. To do this, we can use Bayes' rule to expand $P(s_{t+1}, s_t)$ as:

$$P(s_t|s_{t+1}) = P(s_{t+1}|s_t)\frac{P(s_t)}{P(s_{t+1})} = \frac{P(s_{t+1}, s_t)}{P(s_t)} \frac{P(s_t)}{P(s_{t+1})}$$

then:

$$P(s_{t+1}, s_t) = P(s_t|s_{t+1})P(s_{t+1})$$

Substituting this into the right side of Equation 6, we get:

$$P(s_{t+1}|s_t, \mathcal{F}_t) = \frac{P(s_t|s_{t+1})P(s_{t+1})}{P(s_t)} \tag{7}$$

We can simplify this expression by dividing both the numerator and denominator by $P(s_{t+1})$, which gives:

$$P(s_{t+1}|s_t, \mathcal{F}_t) = \frac{\frac{P(s_t|s_{t+1})P(s_{t+1})}{P(s_{t+1})}}{\frac{P(s_t)}{P(s_{t+1})}} \tag{8}$$

Simplifying the numerator by cancelling out $P(s_{t+1})$ and applying Bayes rules gives:

$$\frac{P(s_t|s_{t+1})P(s_{t+1})}{P(s_t)} = P(s_{t+1}|s_t) \tag{9}$$

Combining Equations 8 and 9 we obtain:

$$P(s_{t+1}|s_t, \mathcal{F}_t) = P(s_{t+1}|s_t) \tag{10}$$

Which is what we wanted to prove in Equation 3. Therefore, we have shown that the Markov property holds for the coalition bargaining game with the introduced filtration.

## 5.6 Implications of the Markovian Embedding

The Markovian embedding of the coalition bargaining game has several implications. First, it allows us to use Markov games theory to analyze the game, which can be useful for understanding the long-term behavior of the game and the equilibrium outcomes. Second, it provides a natural way to simulate the game using MARL methods, which can be used to estimate the equilibrium policies of the involved agents. However, it is important to note that the Markovian embedding is not unique. There are many possible filtrations that can be used to represent the history of the game, and different filtrations may lead to different Markovian processes. Therefore, the choice of filtration can affect the analysis and simulation of the game.

In the context of bargaining games, if we interpret filtrations as *an ordered list of events* we can see how filtrations allow us to order the history of proposals and responses. This can be useful if we want to revisit the older proposals and present them again for the consideration of agents, while avoiding to repeat the newer rejections.

It's worth noting that this approach can be computationally expensive, since we are effectively increasing the dimensionality of the system. Additionally, the choice of memory variables can have a significant impact on the behavior of the resulting Markovian process, and finding an optimal choice of variables can be challenging.

## 5.7   PSEUDOCODE

**Result:** Learn to form coalitions
Initialize an empty Filtration $F$ to record the history of the game;
**for** *each episode* **do**
    Clear the Filtration $F$;
    **while** *unallocated agents exist* **do**
        Select a proposing agent $i$ at random from unallocated agents;
        Agent $i$ proposes a coalition $C$ following its policy;
        Add proposal of coalition $C$ by agent $i$ to Filtration $F$;
        **for** *each agent $j$ in proposed coalition $C$* **do**
            Agent $j$ decides to accept or reject following its policy and observation of Filtration $F$;
            **if** *Agent $j$ accepts* **then**
                Agent $j$ joins the coalition $C$;
                Add acceptance of coalition $C$ by agent $j$ to Filtration $F$;
            **else**
                Add rejection of coalition $C$ by agent $j$ to Filtration $F$;
                Reject the proposal and break;
            **end**
        **end**
        **if** *All agents in proposed coalition $C$ accept* **then**
            Update $Q$ value for all agents in coalition $C$;
        **end**
    **end**
**end**

**Algorithm 1:** A Filtrated Coalitional Bargaining Game

