# OpenReview forum: "MARKOVIAN EMBEDDINGS FOR COALITIONAL BARGAINING GAMES"
_ICLR.cc/2023/TinyPapers — Submitted to Tiny Papers @ ICLR 2023_

### Official Review · Reviewer_u8VJ · 2023-03-22

**Confidence:** 3

**Summary Of Contributions:**

The paper presents a method for converting a non-Markovian coalition game into a Markovian game through the use of filtrations.  This conversion allows existing RL methods to be applied to coalition-based games.  Two short proofs are presented.

**Rating:**

Clear, Correct, and Reproducible (CCR): a submission which meets the reviewing criteria

**Strengths And Weaknesses:**

The paper is relatively clearly written, and is really quite enjoyable to read.  The “findings” are simple, but this is certainly not a bad thing.

I believe the paper is correct, since the proposed method is relatively simple and no empirical methods are presented.  I have outstanding questions about why invoking the relatively complex and off-putting sigma-algebra is strictly necessary.  I feel like one could simply consider an extended state space (with dimension $N\mathbb{C}1 + N\mathbb{C}2 + … N\mathbb{C}N$), and that is somewhat easier to understand.  When I trial a coalition, I deterministically set that extended state to zero.  You present it in terms of the sigma-algebra, and I'm not yet convinced (although I am not an expert on such things) that this is more natural than extending the state space.  Furthermore, how is the sigma-algebra implemented in practice?  If I wanted to implement this in Python, what does this correspond to?

There is an element of “deus ex machina” about the presentation as.  “We cannot do something because it is non-markovian, but, there is a special operator that makes it markovian, and now the games are markovian”.  This is obviously **very** reductionist of me, but it makes me feel like I am misunderstanding the full potential and generality of this work.  I invite the authors to convince me, or, in lieu of a rebuttal, include more discussion as to the practicalities of the filtration operator, and any technical information required to apply it in this domain.

To summarize: The idea of converting a non-Markovian game to a Markovian game, such that existing tools can be applied, is super super appealing, and I think would have substantial impact.  I am just unsure on whether or not the implications of using a filtration operator have been fully fleshed out, and hence the original “research” question actually still applies.  I hope the authors respond to this, because otherwise, it is a very enjoyable TinyPaper.


**Suggested Changes:**

I have some minor suggestions to change the manuscript, which I outline below.

My main request is that the authors disambiguate/discuss how the sigma-algebra relates to the idea of a Markov game on an extended state space, which I referenced above.

Minor:
- Section 2: you provide two inequalities, eg. $| c_1 \cap c_0 | < | c_0 |$.  I don’t understand how either constraint could ever be violated (excluding an equality), and hence I don't understand what it is trying to specify. I could believe that c1 could “add” a node to c0, in which case the cardinalities would be equal, and this would be violated, but i don’t see why that should be excluded as a case.

Typos:
- Pg 1: “...one agent at random propose[s] a coalition…”
- Footnote 1: The “1” typically comes after the period.  Also, the footnote itself should start with a capital letter (“[T]o simplify…”)
- Pg 1: Missing space between “proposal.” and “The”
- Pg 1: Duplicated “The”.
- Throughout: citation style is incorrect.  E.g. the “Okada (1996)” should be formatted as “(Okada, 1996)”, as the work “Okada” should ot feature in the sentence.  For instance, later, “Bachrach et al. (2020)” is correct, because the word “Bachrach” should be in the sentence.
- Pg 1: the sentence containing “... but introduces learning agents _learn_ avoiding …” is poorly worded, to the point where i don’t understand what it is trying to say.
- Throughout: Only proper nouns should be capitalized, eg. “Markov [p]erfect Nash [e]quilibium”.

---

### Official Review · Reviewer_zih4 · 2023-03-28

**Confidence:** 3

**Summary Of Contributions:**

This paper discusses how Coalitional Bargaining Games (CBGs) with non-repeating proposals are non-Markovian in nature and proposes a way to make these CBGs Markovian.

**Rating:**

Clear, Correct, and Reproducible (CCR): a submission which meets the reviewing criteria

**Strengths And Weaknesses:**

S: Well written and good first step.
W: The main text has little technical content

**Suggested Changes:**

1) It would be good to reduce the introduction somewhat and include a few more technical details in the main paper. Including Figures 1 and 2  in the main paper would also allow readers from other domains to understand the proposed concepts better.
2) Typo in Section 1: "The the state dynamics are determined by the agent’s preferences and the rewards are assigned once a proposed coalition is accepted."

---

### Comment · Area_Chair_hfCF · 2023-06-02
**Ready for Archival**

This work meets the threshold for archival, contents the URM statement and is deanonymized.

Congratulations, and good work!

AC hfCF

---

### Meta-Review · Area_Chair_hfCF · 2023-04-05

**Recommendation:** Invite to present
**Confidence:** 4

**Metareview:**

This paper discusses a method for converting non-Markovian games into Markovian games, such that existing solution methods can be applied directly.  Both reviewers noted that this is an appealing idea, and that the solution is, on the surface at least, well-founded.  The idea appears to be novel, and could prove to be significant.  The paper itself is fairly well written, although could benefit from some clarifications highlighted by both reviewers.

The reviews and my own reading of the paper suggest that this paper would benefit from some revisions prior to being released.  However, the work itself shows promise, meets the guidelines of the workshop, and could greatly benefit from exposure and feedback from the community at this early stage.  I therefore recommend that the paper is included.

**Summary:**

This paper discusses converting non-Markovian games into Markovian games through the use of filtrations.  Some theoretical analysis is presented, although no experimental or implementation details are provided.

**Comments And Feedback To The Authors:**

Please take on board the comments of the reviewers and update the manuscript accordingly.  A simple worked example of how a filtration is applied to even a two- or three-player game (in the supplement) would make this paper immediately more impactful and would really help the reader understand the core ideas.  Simulations would then help cement the complexity of the method.

I don't believe actually applying RL to the scenario is required at this stage, but should be the first item on the authors ToDo list when building this work going forwards.

**Reason For Not Giving A Higher Recommendation:**

The absence of empirical findings weakens the impact and generality of the paper.

**Reason For Not Giving A Lower Recommendation:**

I believe the core idea is sound, the work shows promise, and the manuscript is well-prepared.  I believe the authors would benefit from community feedback as well in how to shape the work going forwards, that members of the community would find the work interesting, and that it could engender multiple follow-up works.

---

### Decision · Program_Chairs · 2023-04-07

Invite to present